# Diff-LfD: Contact-aware Model-based Learning from Visual Demonstration for Robotic Manipulation via Differentiable Physics-based Simulation and Rendering

**Xinghao Zhu**[1]    **Jinghan Ke**[2]    **Zhixuan Xu**[3]    **Zhixin Sun**[4]    **Bizhe Bai**[5]    **Jun Lv**[6]

**Qingtao Liu**[3]    **Yuwei Zeng**[7]    **Qi Ye**[3]    **Cewu Lu**[6]    **Masayoshi Tomizuka**[1]    **Lin Shao**[7]

[1]UC Berkeley, [2]USTC, [3]Zhejiang University, [4]Nanjing University

[5]University of Queensland, [6]Shanghai Jiaotong University, [7]National University of Singapore

**Abstract:** Learning from Demonstration (LfD) is an efficient technique for robots to acquire new skills through expert observation, significantly mitigating the need for laborious manual reward function design. This paper introduces a novel framework for model-based LfD in the context of robotic manipulation. Our proposed pipeline is underpinned by two primary components: *self-supervised pose and shape estimation* and *contact sequence generation*. The former utilizes differentiable rendering to estimate object poses and shapes from demonstration videos, while the latter iteratively optimizes contact points and forces using differentiable simulation, consequently effectuating object transformations. Empirical evidence demonstrates the efficacy of our LfD pipeline in acquiring manipulation actions from human demonstrations. Complementary to this, ablation studies focusing on object tracking and contact sequence inference underscore the robustness and efficiency of our approach in generating long-horizon manipulation actions, even amidst environmental noise. Validation of our results extends to real-world deployment of the proposed pipeline. Supplementary materials and videos are available on our webpage: https://sites.google.com/view/diff-lfd.

**Keywords:** Learning from Visual Demonstration, Model-based Robotic Manipulation, Differentiable Physics-based Simulation and Rendering

## 1   Introduction

Learning from Demonstration (LfD) empowers robots to acquire policies from expert demonstrations, such as those available on YouTube [1], which can reduce the human effort involved in robotic skill learning [2, 3]. This paper delves into the development of a model-based LfD pipeline that employs raw RGB videos as inputs. While model-based learning approaches have been acknowledged for their potential for superior sample-efficiency and generalization compared to model-free approaches [4–6], model-based LfD remains under-explored. Several major challenges hinder the wide application of model-based LfD in the physical world.

One challenge is how to automatically and efficiently develop a model that scales to high-dimensional input such as raw images or videos [7]. To tackle this, we introduce a self-supervised modeling pipeline that leverages recent advancements in differentiable rendering and signed distance functions. This pipeline estimates both the geometric shape of the object and its associated 6D poses, forming an explicit representation. A second challenge lies in enabling robots to effectively utilize physical models to generate efficient policies. This is particularly critical for robots operating in real-world contact-rich manipulation tasks where the physical interaction between the robot and its environment is a key factor [8, 9]. To address this, we develop a hierarchical LfD framework that integrates low-level modules for contact-point localization and contact-force optimization with

---

Correspondence to zhuxh@berkeley.edu and linshao@nus.edu.sg

7th Conference on Robot Learning (CoRL 2023), Atlanta, USA.

a high-level module for contact sequence planning. These modules work in concert to plan manipulative actions. To ensure robust and real-time deployment, we further incorporate a neural policy designed to imitate the outcomes of planning algorithms. This enables the robot to execute complex tasks with high reliability and efficiency.

We have evaluated our pipeline on two datasets, including the sth-sth dataset [10] containing basic manipulation actions on various objects and a small recorded video dataset showing a human performing dexterous in-hand manipulation with primitive objects. The results, derived from rigorous simulation and real-world experiments, bear testament to the effectiveness of our proposed pipeline.

Our key contributions can be summarized as follows: 1) introduce a novel framework for model-based learning from visual demonstrations, 2) provide a self-supervised approach for pose estimation and shape reconstruction, utilizing the differentiable rendering, 3) develop a hierarchical policy that combines the low-level contact-point localization and contact-force optimization based on the differentiable simulation and high-level contact sequence planning, with neural imitation learning for efficient and robust real-world execution, 4) conduct comprehensive experimental validation of our algorithms in both simulated and real-world environments to demonstrate their efficacy and robustness.

## 2 Diff-LfD Framework

**Overview.** Given a demonstrated RGB video consisting of $N$ frames denoted as $\mathcal{V} = \{\mathcal{I}_t\}_{t=1}^N$, we prepossess the video to segment and identify the most relevant objects with masks $\{\mathcal{M}_t\}_{t=1}^N$, exploiting the SAM [11]. The local frame of the object is randomly defined at the first frame. Our *Diff-LfD* calculates the object's relative pose transformation in the demonstration and jointly estimates the object's mesh $\mathcal{O}$ and the associated 6D poses $\{\mathcal{P}_t\}_{t=1}^N$ at each frame. If the robot is provided with a similar but different object from the object recorded in the video, we align the pose of the provided manipulated object with the reconstructed object. Next, our pipeline infers the *wrench* (a combination of external forces and torques) required to complete the pose transformation across two consecutive time steps and generates feasible robot actions to accomplish the pose transformation. This planning includes both the low-level contact-point localization and contact-force optimization and high-level contact sequence planning to chain the whole (long-horizon) manipulation sequences. The manipulation actions are then utilized to train a neural network for robust real-world execution and generalization. We provide an explanation of the pipeline, *wrench*, and object alignment in Appendix.

### 2.1 Pose and Shape Estimation with Differentiable SDF

This subsection introduces the pipeline for pose estimation and shape reconstruction from raw videos. We adopt the differentiable SDF (*Diff-SDF*) [12] to represent the object geometry, which has a large representation capability to model diverse objects with various topology structures. Moreover, Diff-SDF enables smooth image-based optimization to the images due to its inherent convexity [12]. The explicit surface mesh $\mathcal{O}$ can be extracted from the SDF using the marching cube method [13].

Ideally, given an initialization of an SDF parameterized by $\phi$ and its associated 6D poses at each frame $\{\mathcal{P}_t\}_{t=1}^N$, the differentiable renderer $\mathcal{R}$ produces a sequence of images $\{\mathcal{I}(\phi, \mathcal{P}_t)_t\}_{t=1}^N = \{\mathcal{R}(\phi, \mathcal{P}_t)\}_{t=1}^N$. *Diff-SDF* optimizes the SDF parameters to reconstruct the object shape by reducing the reconstruction loss:

$$\mathcal{L}^R = \sum_{t=1}^N ||\mathcal{I}(\phi, \mathcal{P}_t)_t - \mathcal{I}_t|| \tag{1}$$

However, this approach encounters difficulties when applied to real-world videos due to the following reasons: *unknown camera poses* and *lack of views for unseen regions*. Because the SDF optimization presumes that camera poses of $\mathcal{I}_t$ are known in advance, which is not valid for real-world videos where camera poses are not provided. To estimate camera poses $\{\mathcal{P}_t^{-1}\}_{t=1}^N$, we employ differentiable rendering to hierarchically produce an explicit surface mesh $\hat{\mathcal{O}}$ with texture denoted

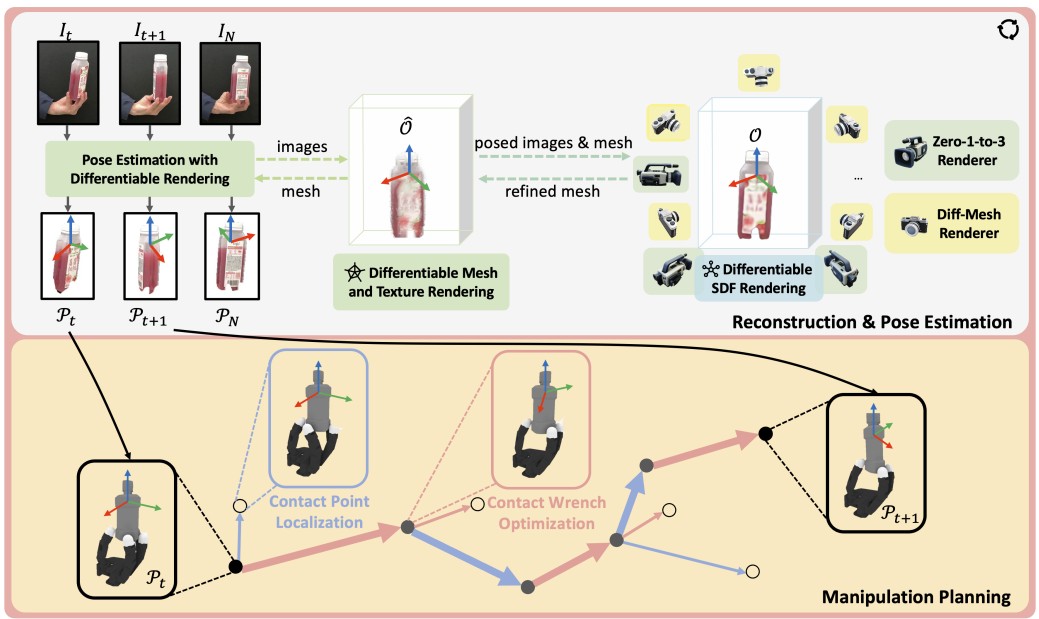

Figure 1: The proposed model-based learning from demonstration (LfD) pipeline can be divided into two primary components. The top part focuses on object shape reconstruction and pose estimation, employing differentiable mesh rendering and signed distance function (SDF) (Section 2.1). The bottom part illustrates the process of contact-aware hierarchical manipulation planning, involving contact point localization and differentiable wrench optimization (Section 2.2).

as $\hat{\mathcal{T}}$, and jointly estimate the objects poses $\{\hat{\mathcal{P}}_t\}_{t=1}^N$ over multiple images. Details are provided in the Appendix.

We denote the optimized mesh, textures, and poses from differentiable rendering as $\hat{\mathcal{O}}^*, \hat{\mathcal{T}}^*, \{\hat{\mathcal{P}}_t^*\}$, respectively, and their associated rendered images $\hat{\mathcal{I}}_t^* = \mathcal{R}(\hat{\mathcal{O}}^*, \hat{\mathcal{T}}^*, \{\hat{\mathcal{P}}_t^*\})$ as:

$$\hat{\mathcal{O}}^*, \hat{\mathcal{T}}^*, \{\hat{\mathcal{P}}_t^*\} = \underset{\hat{\mathcal{O}}, \hat{\mathcal{T}}, \{\hat{\mathcal{P}}_t\}}{\arg\min} \sum_{t=1}^N \|\hat{\mathcal{I}}_t - \mathcal{I}_t\| \quad \text{where} \quad \hat{\mathcal{I}}_t = \mathcal{R}(\hat{\mathcal{O}}, \hat{\mathcal{T}}, \{\hat{\mathcal{P}}_t\}) \tag{2}$$

The quality of mesh $\hat{\mathcal{O}}^*$ is usually not satisfying. We then optimize the Diff-SDF to get an optimized SDF $\phi^*$ by setting the camera pose to be $\{\hat{\mathcal{P}}_t^{-*}\}$ to reduce the projection loss:

$$\phi^* = \underset{\phi}{\arg\min} \sum_{t=1}^N \|\mathcal{I}(\phi, \hat{\mathcal{P}}_t^*)_t - \hat{\mathcal{I}}_t^*\| \quad \text{where} \quad \hat{\mathcal{I}}_t^* = \mathcal{R}(\hat{\mathcal{O}}^*, \hat{\mathcal{T}}^*, \{\hat{\mathcal{P}}_t^*\}) \tag{3}$$

After the Diff-SDF optimization, the resulting surface mesh $\hat{\mathcal{O}}^{**}$ is then extracted from the SDF $\phi^*$. The $\hat{\mathcal{O}}^{**}$ is then leveraged to optimize the object poses $\{\hat{\mathcal{P}}_t^{**}\}$ as below. The process in Eqn 2-4 iterates until we get a small loss below a given threshold or reach the maximum iteration number.

$$\{\hat{\mathcal{P}}_t^{**}\}, \hat{\mathcal{T}}_t^{**} = \underset{\{\hat{\mathcal{P}}_t\}, \hat{\mathcal{T}}}{\arg\min} \sum_{t=1}^N \|\hat{\mathcal{I}}_t - \mathcal{I}_t\| \quad \text{where} \quad \hat{\mathcal{I}}_t = \mathcal{R}(\hat{\mathcal{O}}^{**}, \hat{\mathcal{T}}, \{\hat{\mathcal{P}}_t\}) \tag{4}$$

Although each video contains multiple frames, there are still cases that lack sufficient views, resulting in poorly reconstructed unseen regions of the object. To address the incomplete views, we adopt a diffusion model [14] to infer the unseen areas. Our model takes the first real image $\mathcal{I}_1$ with a known camera pose and synthesizes images from different viewpoints around the object. We then combine these synthetic views with others for a complete object-shape reconstruction.

## 2.2 Contact-Aware Manipulation Policy

Building on the estimated object's pose and shape, this subsection delves into the process of manipulating an object between two consecutive poses $\mathcal{P}_t$ and $\mathcal{P}_{t+1}$. If the robot is provided with a

similar but different object from the object recorded in the video, we align the pose of the provided manipulated object with the reconstructed object, with details in Appendix.

Our framework employs a hierarchical structure consisting of low-level modules for contact-point localization and contact-force optimization, as well as high-level contact sequence planning. The low-level modules serve dual purposes: contact-point localization allows the robot to establish new contacts while keeping the object stationary, whereas contact-force optimization enables the robot to manipulate the object toward its target and maintain stable contact. These low-level actions are then orchestrated by the high-level contact sequence planning module to form a cohesive sequence of actions. To facilitate efficient and robust real-time deployment, we also incorporate a neural policy designed to imitate the planned trajectories.

**Contact point localization.** Contact point localization enables the robot to change contact points on the object, which encompasses two critical steps: *the generation of the transition target* and *the execution of the transition*. The transition target is calculated analytically with the desired object transformation wrench $\mathcal{W}$. Wrench $\mathcal{W}$ is located at the objects' center and represents the necessary wrench to facilitate the transformation from $\mathcal{P}_t$ to $\mathcal{P}_{t+1}$. It is determined using a Proportional-Derivative (PD) controller: $\mathcal{W} = k_p * (\mathcal{P}_{t+1} - \mathcal{P}_t) - k_d \dot{\mathcal{P}}_t + g$, where $k_p, k_d$ are the proportional and derivative gains, and $g$ signifies the gravitational and external forces acting upon the object. Following this, we use an enumeration process to identify all plausible contact combinations that can generate the desired wrench $\mathcal{W}$. Initially, all potential contacts are assessed to single out those capable of producing the desired object wrench $\mathcal{W}$ through contact points $\{p_i\}$ for $i \in [1..n]$. The number of contact points $n$ is pre-determined based on the manipulation task. Further filtering processes are implemented to ascertain contacts that meet kinematic and stationary constraints: the inverse kinematics are solved to verify kinematic feasibility, and only one contact point can move at a given time while the remaining contact points hold the object immobile. Although multiple contact points could theoretically move while maintaining the object stationary, we found that planning is considerably more complex due to the enlarged search space, and the objects are prone to unintended movement due to execution errors. To execute the transition, we use the Rapidly-exploring Random Tree (RRT) motion planner to generate a feasible trajectory for the moving contact and the gravity compensation wrench on the remaining contacts to hold the object. More details of the contact point localization are provided in Appendix.

**Contact wrench optimization.** Once the contact points $p = \{p_i\}$ are determined, the robot exerts contact wrenches $\mathcal{W}^p = \{\mathcal{W}^{p_i}\}$ at contact points to manipulate the object toward its target. The objective and loss functions to optimize the contact wrenches are defined in Eq. 5.

$$\min_{\mathcal{W}^p} \mathcal{L}(\mathcal{W}^p) = \lambda_{\mathcal{P}} \|\mathcal{P}' \ominus \mathcal{P}_{t+1}\| + \lambda_v \left\|\dot{\mathcal{P}}'\right\| + \lambda_{\mathcal{W}} \|\mathcal{W}^p\| \quad \text{subject to} \quad \mathcal{P}' = \mathcal{F}(\mathcal{P}, \mathcal{W}^p) \quad (5)$$

where $\mathcal{F}$ represents the contact dynamics, $\mathcal{P}'$ is the 6D object pose after applying the wrench $\mathcal{W}^p$ from the initial object pose $\mathcal{P}$. $\mathcal{P}_{t+1}$ is the target object pose, $\ominus$ represents subtraction for 6D poses. $\dot{\mathcal{P}}'$ represents the object's velocity and is added to damp the object's speed, making the manipulation more stable [15]. $\lambda_{\mathcal{P}}, \lambda_v, \lambda_{\mathcal{W}}$ are hyperparameters standing for loss weights.

We propose using gradient-based methods to optimize the objective function, Eq. 5, with differentiable simulation in Nimble Physics[16] to approximate the forward dynamics $\mathcal{F}$. We use $s = (\mathcal{P}, p)$ as the concatenated state of the system in the following of this paper. The gradient of the objective can be computed as $\nabla = \frac{\partial \mathcal{L}}{\partial \mathcal{W}^p}$ from the simulation. However, gradients near the contact are often nonlinear, sensitive, and discontinuous, posing challenges for vanilla gradient descent optimization methods. To address this issue, this paper draws inspiration from [17–20] and proposes computing the gradient expectation at each point with Gaussian noises, as shown in Eq. 6. The contact wrench is then updated using a step size $\alpha$ along the gradient direction. In this paper, we use the analytical contact wrench computed during the contact point localization as the initial solution point for the optimization process.

$$\nabla = \mathbb{E}_{n_s, n_{\mathcal{W}} \sim \mathcal{N}} \left[ \frac{\partial \mathcal{L}(\mathcal{F}(s + n_s, \mathcal{W}^p + n_{\mathcal{W}}), \mathcal{W}^p + n_{\mathcal{W}})}{\partial \mathcal{W}^p} \right] \quad (6)$$

**High-level planning.** Our global contact sequence planning, as detailed in Algorithm 1, employs hierarchical planning to identify viable manipulation sequences, utilizing the previously introduced contact point localization (`ContactLoc`) and contact wrench optimization (`OptWrench`). While the exertion of contact wrenches allows the robot to perform manipulation tasks involving nearby target poses, switching between multiple contacts is necessary when dealing with distant targets due to kinematic limitations.

Every node $s$ in the planning tree encapsulates the object pose $\mathcal{P}$ and contact $p$. The tree begins with a start node that represents the initial object pose and robot contact, with the goal

---

**Algorithm 1** Global Planning for Manipulation Sequences

1: **Input:** $s_0 = (\mathcal{P}_t, p_t)$, target object pose $\mathcal{P}_{t+1}$
2: **Output:** $\mathcal{R} = \{s\}$
3: $\mathcal{Q} \leftarrow \{s_0\}, \mathcal{R} \leftarrow \{\varnothing\}$       ▷ Init.
4: **while** $\mathcal{Q}$ is not empty **do**
5:   $s \leftarrow \texttt{SelectNode}(\mathcal{Q})$
6:   **if** $\texttt{IsSuccess}(s, \mathcal{P}_{t+1})$ **then**
7:    Return $\mathcal{R}$      ▷ Exit if success
8:   **end if**
9:   $s' \leftarrow \texttt{OptWrench}(s, \mathcal{P}_{t+1})$    ▷ Opt. wrench
10:   **if** $\texttt{OptIsSuccess}(s, s')$ **then**
11:    $\mathcal{Q} \leftarrow \mathcal{Q} \cup s'; \mathcal{R} \leftarrow \mathcal{R} \cup s'$
12:   **else**
13:    $\mathcal{S} \leftarrow \texttt{ContactLoc}(s, \mathcal{P}_{t+1})$   ▷ Loc. contacts
14:    **for** $s' \in \mathcal{S}$ **do**
15:     $\mathcal{Q} \leftarrow \mathcal{Q} \cup s', \mathcal{R} \leftarrow \mathcal{R} \cup s'$
16:    **end for**
17:   **end if**
18: **end while**
19: Return $\mathcal{R}$

---

of reaching the target object pose $\mathcal{P}_{t+1}$. At each iteration, a node $s$ is chosen and expanded using `ContactLoc` or `OptWrench` following A* search [21]. If the object has been successfully manipulated through the exertion of an optimized contact wrench (`OptIsSuccess`), the resulting node $s'$, containing the manipulated object pose and contact points, is expanded. Conversely, if the exertion fails, contact localization is performed by identifying a new set of contacts and expanding them in the tree. This planning algorithm continues until either the target object pose is reached (`IsSuccess`) or all nodes within the tree have been explored. The search procedure we propose focuses on optimizing contact wrenches at first and resorts to locating new contacts only if the exertion fails. Although this approach narrows down the search space, it may also prune potentially valid and optimal paths. For instance, transiting contacts before reaching the kinematic limit might result in a shorter trajectory with fewer contact switches. To address this, we introduce a random chance for each node to transition contacts, regardless of the wrench optimization outcome. This design promotes exploration within the planning process, enabling a more comprehensive discovery of the entire search space.

**Sim2Real: closed-loop policy with domain randomization.** Despite its efficacy in generating viable manipulation trajectories, the above-described planning algorithm is computationally demanding as it requires online enumeration of contacts and wrench optimization, rendering it unsuitable for real-time applications. To surmount this challenge, we utilize deep learning to approximate the manipulation policy. We leverage a fully connected network to learn the robot control commands that were derived from the high-level planning algorithm. This network ingests the object pose and joint angles as inputs and outputs of robot joint torques. These torques are obtained by mapping the contact wrenches into joint torques, courtesy of the Jacobian. Our training dataset is generated by solving the planning problem under conditions of noisy initial and target positions and perturbed system dynamics. This process results in a set of state-torque training pairs. We further augment each sample by introducing noise into the states and optimizing the joint torques to adhere to the planned trajectory. It's noteworthy that for domain randomization, we optimize the contact wrench to reach the next state along the trajectory rather than solving the original planning problem with a distant target. This makes the data augmentation process more efficient. In a bid to further enhance performance, we fine-tune the network within a Markov Decision Process (MDP) framework using the REINFORCE algorithm [22]. During the fine-tuning process, the state and action spaces maintain the same setup as described earlier, while the reward function is defined as $r(s, \tau) = -\lambda_{\mathcal{P}} \| s \ominus \mathcal{P}_{t+1}\| - \lambda_v \|\dot{s}\| - \lambda_{\mathcal{W}} \|\tau\|$, where $\mathcal{P}_{t+1}$ is the target object pose, and $\dot{s}$ denotes the object velocity [15]. $\lambda_{\mathcal{P}}, \lambda_v, \lambda_\tau$ are hyperparameters.

| | Pull Right | Pull Left | Push Right | Push Left |
|---|---|---|---|---|
| **Baseline [23]** | 0.976 | 0.992 | 0.994 | 0.946 |
| **Ours** | 1.000 | 1.000 | 1.000 | 1.000 |

Figure 3: Baseline comparisons on LfD framework. Each cell represents the success rate of the manipulation.

| | RRT | | CITO | | PGDM | | iLQR | | Ours | |
|---|---|---|---|---|---|---|---|---|---|---|
| **Ball** | $122 \pm 20$ | $7.2°$ | $52 \pm 8$ | $16.7°$ | $2.14 \pm 0.4$ | $2.4°$ | $57 \pm 10$ | $11.2°$ | $62 \pm 12$ | $2.6°$ |
| **Cube** | $136 \pm 16$ | $9.0°$ | $60 \pm 7$ | $18.5°$ | $2.16 \pm 0.3$ | $4.1°$ | $70 \pm 19$ | $13.3°$ | $78 \pm 16$ | $3.8°$ |
| **Capsule** | $127 \pm 24$ | $8.4°$ | $63 \pm 4$ | $15.2°$ | $2.18 \pm 0.3$ | $9.3°$ | $80 \pm 6$ | $12.2°$ | $81 \pm 8$ | $7.7°$ |

Figure 2: Experimental results from sth-sth (1st & 2nd rows) and in-hand object manipulation (3rd & 4th rows).

Figure 4: Baseline comparisons on contact-aware manipulation policy. The first element in each cell is the mean/variance for the computation time ($s$); the second is the difference between the target and final object rotation ($°$).

# 3 Experiments

This section offers both quantitative and qualitative assessments of our proposed methodology. Our experiments are designed to address the following research questions: 1) How does our Diff-LfD framework compare to baselines that also rely on visual demonstrations? 2) What is the efficacy of our contact-aware manipulation algorithm in generating long-horizon trajectories? 3) Is our approach feasible for deployment in real-world scenarios? 4) How accurate is our self-supervised object reconstruction and tracking? 5) What is the utility of the views synthesized by the diffusion model? 6) What impact do gradient-based optimization, global planning, and random contact transition have on performance? 7) How robust are the generated trajectories and the closed-loop policy? We conducted evaluations in two distinct experimental settings: basic manipulation tasks involving primitive objects and more complex in-hand object manipulation tasks.

Experimental setups and ablation studies addressing questions 4-7 are elaborated in Appendix.

**Baseline comparisons on LfD framework.** We compare our model-based approach with the method introduced in [23]. Petrík et al. [23] presents an optimization-based method to estimate a coarse 3D state representation, using a cylinder for the hand and a cuboid for the manipulated object(s). Such coarse approximation limits the representation capability and the quality of the state estimation. We utilize our object reconstruction and tracking to estimate the object trajectory and use contact planning to find a path. We select videos of 4 classes from the sth-sth dataset [10]: "Pull Right" with 164 videos, "Pull Left" with 130 videos, "Push Right" with 89 videos, and "Rush Left" with 253 videos. We report the results as in Fig. 3. Our approach successfully finished all the classes and slightly outperformed the method introduced in [23]. One explanation is that these four types of videos are simple for our proposed pipeline to imitate. Thus, we also apply our method for in-hand manipulation tasks from raw videos to test the limits of our proposed framework. Fig. 2 shows the manipulation trajectory in two environments generated by our method.

**Baseline comparisons on shape reconstruction and pose estimation.** In contrast to other learning-based approaches for shape reconstruction and pose estimation, such as Neural Radiance Fields (NeRF), our perception module operates under a distinct task setting. Specifically, our input consists of a single-object RGB video featuring objects that undergo both rotation and translation. Most NeRF-based methods, on the other hand, rely on multiple static object poses with known camera positions. Some NeRF implementations utilize COLMAP [24] to initialize camera poses. However, this approach is less effective in our setting, where the background remains largely unchanged, and the frame count is limited. These factors hinder COLMAP's ability to accurately estimate object poses, leading to unstable object surface reconstructions from NeRF. Further experimental comparisons with Nope-NeRF [25], which also employs COLMAP for initialization, are available in the webpage. Our findings indicate that Nope-NeRF fails to converge in more than half of the test cases (5 out of 9), resulting in empty reconstructions. The remaining cases yielded incorrect pose estimations and reconstructions when compared to our method.

**Baseline comparisons on the contact-aware manipulation policy.** In this study, we evaluate our contact-aware trajectory planning algorithm against four established baselines within the context of in-hand object rotation tasks. The baselines are as follows: 1) The Rapidly-Exploring Random Tree (RRT) planner, as outlined in [17], employs random sampling within a configuration space defined by both robot joint positions and object poses to identify feasible trajectories. 2) Contact-Implicit Trajectory Optimization (CITO) [26] first establishes a predefined trajectory for the object, then identifies optimal contact points along this path before calculating the requisite control inputs for trajectory tracking. 3) Pre-Grasp Informed Dexterous Manipulation (PGDM) [27] utilizes reinforcement learning to train manipulation agents, incorporating pre-computed grasp data to achieve the desired manipulation trajectory. 4) The Iterative Linear Quadratic Regulator (iLQR) [17] employs local approximations of the dynamical system to iteratively solve for optimal manipulation strategies through quadratic planning. For the purposes of this experiment, our algorithm operates without the closed-loop policy detailed in Sec. 2.2. We apply baselines on three in-hand manipulation tasks associated with the ball, the cube, and the capsule. We adopt two evaluation metrics: the averaging planning time and the difference between the target rotation and the final object rotation. Results are reported in Fig. 4.

While both PGDM and iLQR boast the quickest inference times, it's crucial to highlight that PGDM requires approximately 5 hours of training for each task, and iLQR suffers from a higher tracking error compared to our method. The RRT approach uniformly expands its search tree, thereby increasing the probability of encountering unstable contacts and consequently requiring the most time to complete the task. In contrast, our algorithm and CITO focus on a more constrained search space where stable grasping is feasible, thereby simplifying the search complexity. Furthermore, our empirical results indicate that the final error rates for all baseline methods were consistently higher than our algorithm. Specifically, the RRT approach lacks a guarantee for optimal trajectory sampling, CITO overlooks physical dynamics during the planning phase, and iLQR struggles with optimization over nonlinear loss contours. These limitations render the baseline methods susceptible to failure due to dynamic uncertainties and execution errors.

**Real-world experiments.** We conducted real-world experiments for in-hand object manipulation. The experimental setups are illustrated in Fig. 5. We trained the closed-loop policy as discussed in Sec. 2.2 to imitate a human rotating a cube and deployed it as the robot controller. Experimental videos are available in the webpage. This network receives the current joint angles of the robot, and the current object poses as input and outputs the joint torques. We performed the in-hand manipulation task with the Allegro Hand for different primitive objects and initial poses. During supervised learning, convergence of the policy is achieved in approximately 9.3 minutes, while fine-tuning takes an average of 50.1 minutes. The results of the difference between the target and final object rotation errors are Cylinder (3.8°), Ball (2.4°), Lemon (6.3°), and Avocado (5.9°), which further underscore the ability of our closed-loop policy to generalize across similar but distinct geometries.

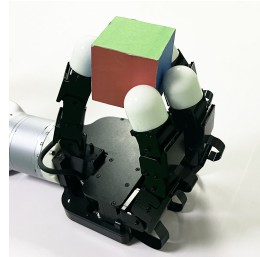

Figure 5: Allegro Hand performs in-hand object manipulations.

Additional experiments covering topics such as object reconstruction and tracking, the efficacy of the diffusion model, gradient-based optimization techniques, random contact transitions, and robustness analyses are provided in Appendix.

## 4 Related Work

**Learning from visual demonstration.** We focus our review on the approaches that utilize visual data or adopt a model-based approach. For a broader review, we refer readers to [28]. One line of work in LfD [29–35] learns the cost and reward function from visual demonstrations. To extract knowledge from images/videos, various works [36–42] adopt representation-learning approaches to distill low-dimensional latent states and action representations. Developing explicit representation for LfD has received little attention due to the limited representation capability. Petrík et al. [23]

adopt coarse 3D cylinders and cuboids to present the hand and manipulated objects. To tackle the representation capability issue, we propose a self-supervised modeling pipeline that estimates the fine-level geometric shape of the object and its associated 6D pose sequences from raw videos. Moreover, our approach infers the contact forces underlie these poses, contributing to model-based learning algorithms.

**Differentiable simulation and rendering in robotic manipulation.** Recently, great progress has been made in the field of differentiable physics-based simulation and differentiable rendering [43–58], which for a broader review, please refer to [59, 60]. These differentiable tools have been applied in robotic manipulation tasks [20, 61–69]. We use the differentiable physics simulation to optimize the contact forces for in-hand manipulation tasks and proposed an iterative pose estimation and shape reconstruction pipeline from raw RGB videos via the differentiable rendering [70] and differentiable signed distance functions [12].

**Model-based manipulation.** The use of contact dynamics often leads to non-convex optimization problems, causing difficulties in finding local optima due to the discontinuity introduced by contact switching [19, 71, 72]. Contact-implicit trajectory optimization (CITO) [73–76] addresses this issue by planning manipulation actions without a pre-specified contact schedule. Chen et al. [26] further considers finger gaiting primitive in trajectory planning but assumes reachable states and pre-defined object trajectories, leading to potential failures due to ignorance of dynamical restrictions and control errors. Pang et al. [17] uses a convex quasi-dynamics model with a rapidly exploring random tree (RRT) to directly search feasible manipulation actions at the dynamical level, although the manipulated object lies on a surface and doesn't require consideration of gravity. Our work plans manipulation actions directly at the dynamical level to address system noises and utilizes differentiable physics simulations for contact optimization and contact localization for efficient search.

## 5   Limitation

For the object shape reconstruction and pose estimation, we assume that the RGB videos are segmented, and the majority of the mass is concentrated as its geometry center. Our pipeline currently works only with rigid bodies, not with articulated rigid bodies or deformable objects. Although we leverage the diffusion model to mitigate the occlusion and reduce shape reconstruction uncertainty, the quality varies for real RGB videos when testing our pipeline in the wild. For contact-aware manipulation, we focus on tasks that require the object to move along a demonstrated trajectory. Generalizing the algorithm to other tasks with sparse rewards will be left for future work. Our approach relies on the differentiable physics-based simulation to generate the contact wrench with domain randomization to reduce the sim2real gap. Complicated physics interaction might fail to be captured by the differentiable physics simulation. We are interested in adding residual/learning layers to augment the differentiable physics simulation to align with the real world in future work.

## 6   Conclusion

This paper investigates the use of model-based learning from demonstrations for robotic manipulation tasks, contributing several significant aspects to the field. First, we introduce a new framework for learning from human visual demonstrations in a self-supervision manner, which has the potential to generate robot skills at a large scale. Second, we utilize differentiable rendering to track object poses in a self-supervised manner. Third, we design a high-level planning framework that employs differentiable simulations to generate long-horizon contact actions. This includes inferring and transitioning contact points, optimizing contact forces, and exerting them. The manipulation trajectories are then approximated by a neural network. Finally, we conduct experiments to evaluate the effectiveness of our approach from multiple angles. Our results demonstrate the robustness and efficiency of our proposed method to learn from human demonstrations and outperform existing approaches by a large margin.

**Acknowledgments**

Xinghao Zhu is supported by the UCB-FANUC Fellowship. This work was in part supported by a startup grant from the National University of Singapore.

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
