# OpenReview forum: "Diff-LfD: Contact-aware Model-based Learning from Visual Demonstration for Robotic Manipulation via Differentiable Physics-based Simulation and Rendering"
_robot-learning.org/CoRL/2023/Conference — CoRL 2023 Oral_

### Official Review · Reviewer_jZUA · 2023-07-05

**Confidence:** 3
**Originality:** Good
**Technical Quality:** Good
**Clarity Of Presentation:** Fair
**Impact:** 3

**Recommendation:**

Weak Reject: I recommend rejecting the paper, but will not argue for my recommendation if the majority of other reviewers have a different opinion.

**Review:**

### Strengths

- This work clearly required a substantial amount of engineering effort to integrate multiple systems (differentiable rendering for object reconstruction, planning contact locations, etc.).
- The paper includes hardware demonstrations.

### Weaknesses

- At a number of points, the paper is unclear and defers important details to the supplementary materials. The paper should be able to clearly explain the essence of your approach without reference to the supplementary materials.
- The reported experiments are very simple, mostly block pushing and in-hand reorientation of cubes and spheres.

### Major comments

1. Section 2.1 is unclear. Are you working with SDFs or with explicit meshes, or both? Consider revising this section to improve clarity.

**Quality Of The Limitations Section:**

Limitations are addressed clearly

**Questions For Rebuttal:**

In addition to issues identified above, please address the following:

1. How is the diffusion model in section 2.1 trained?
2. What is the quality of the meshes that result from the differentiable rendering reconstruction process? Are the normals accurate? I would imagine that that would be important for accurate simulation.
3. You mention that fine-tuning your neural network with REINFORCE was helpful. Did you compare with a neural network that was trained end-to-end with RL?
4. How well will this approach generalize to more challenging manipulation tasks? For example, many manipulation tasks (e.g. pouring, insertion) are not simply about minimizing the pose error.
5. How long does your approach take to train on new objects? How much time is spent on object reconstruction, planning, and fine-tuning the neural network?
6. How are the contact points chosen? Are they manually specified for each object, or are they somehow learned from the reconstruction?

**Update after discussion**

This paper seems like a well-implemented systems paper; if CoRL had a track for systems papers (like RSS), then this would be a clear accept. As it stands, it is still unclear to me where the contribution is. The authors attempted to clarify their contribution in their rebuttal, but seem to have just restated the contributions provided in their paper without clarifying the specific advances over prior work. For example, differentiable rendering has been used for pose estimation in prior works; what is new about the differentiable rendering sub-system in this paper? If one of the other reviewers has a clear sense of the contribution, then I will defer to them, but as it stands I don't feel confident increasing my score.

**Robotics Focus:**

Sufficient demonstration on hardware

**Summary Of Paper:**

This paper proposes a pipeline for in-hand manipulation tasks that involves first using a differentiable renderer to reconstruct the object shape and pose, then using a gradient-based contact planner (combined with search over contact configurations) to find sequences of actions that move the object to a desired pose. The planner is then distilled into a neural network control policy that is demonstrated on hardware experiments.

**Summary Of Recommendation:**

Each of the subsystems proposed in this paper seem fairly incremental (e.g. differentiable rendering for object pose reconstruction, gradient-based planning of contact wrenches). Certain elements are interesting (combining gradient-based planning with search) but are not sufficiently explored in the paper and contain a number of heuristics that are not well justified via ablations (e.g. the random chance of transitioning contacts). If the paper were revised to clarify the novelty of each subsystem (or the novelty in combining all these subsystems, which is equally valid), as well as answering my other questions above, I would consider upgrading my recommendation.

---

> ### Author Response · Authors · 2023-08-13
> **Authors' Response to Reviewer jZUA (Part 1)**
>
> We appreciate the reviewer’s constructive suggestions/questions/comments for further clarifying and improving our work.
>
> **We first address your major concern about the contribution.**
>
> **Our contribution at the systematic level**, we proposed a novel model-based framework for learning from demonstration to empower robots to mimic visual demonstrations. The framework utilizes the “comparing sim-real and then improving” fashion leveraging the differentiable simulation and rendering, thus is in a self-supervision manner, which has the potential to generate robot skills at a big scale.
>
> **We discuss our contribution to each submodule.** **"differentiable rendering for object pose reconstruction."**
> Our problem setting is given an RGB video (without the known camera pose), jointly reconstructing shape and estimating poses for a moving object (with large translation and rotation). This problem setting is extremely challenging. Our proposed approach works for a large variety of object shapes and sufficient accurate pose estimations. We have compared it with other NeRF works and demonstrated the NeRF does not work well in our problem setting.
>
> **"gradient-based planning of contact wrenches"** We proposed to adopt the **contact-centric** hierarchical policy generation pipeline. We compare our method with other planning algorithms (see main paper line 247-268) and show ours outperforms the baselines by a large margin (see main paper line 253 above, figure 4). Specifically, the contact points Inference generates feasible contact configurations by focusing on desired object transitions at the force level. It provides a systematic approach to understanding how objects can be manipulated based on their intended movement paths. The contact wrench optimization utilizes a gradient-based method to infer the optimal force execution. It ensures that the forces applied are both efficient and effective in achieving the desired manipulation. A high-level global planning framework integrates both gradient-based and search-based low-level primitives to create comprehensive manipulation plans.

---

> ### Author Response · Authors · 2023-08-13
> **Authors' Response to Reviewer jZUA (Part 2)**
>
>
> **Below, we address other questions from the reviewer.**
>
> **Some questions are answered in the general response.**
>
> **Q1. The reported experiments are very simple, mostly block pushing and in-hand reorientation of cubes and spheres.**
>
> The reorientation of cubes and spheres might look trivial. However, in-hand manipulation with the dexterous hand is extremely challenging. Small errors in the fingertips might lead to failure cases. Fairly amount of effort has been put in order to run the real robot experiments. The majority of works are rotating objects with primitive geometry, especially demonstrating with the real dexterous hand. To our best knowledge, we are the first ones to demonstrate dexterous manipulation within the learning from a visual demonstration setting.
>
> **Q2. Section 2.1 is unclear. Are you working with SDFs or with, explicit meshes, or both? Consider revising this section to improve clarity.**
>
> When we generate the shape from the image sequences, SDFs are used. Then our pipeline converts SDFs into explicit meshes using the matching cube algorithms. We use the converted explicit meshes to estimate object poses.
>
> **Q3. What is the quality of the meshes that result from the differentiable rendering reconstruction process? Are the normals accurate?**
>
> We have included visual illustrations of the reconstructed mesh, and details can be found in Q2 of the general response. It's important to note that in our algorithm, the normal doesn't need to be precisely accurate. The reconstructed mesh can be aligned with the manipulated object during the contact-aware planning phase and fine-tuned with the closed-loop policy to encounter reconstruction error, allowing for flexibility in the implementation.
>
> **Q4. You mentioned that fine-tuning your neural network with REINFORCE was helpful. Did you compare it with a neural network that was trained end-to-end with RL?**
>
> We have conducted additional experiments to compare our approach with PGDM [1] in in-hand manipulation tasks, details can be found in Q5 of the general response. The PGDM employs pre-grasp planning and PPO for learning manipulation, utilizing a trajectory tracking loss akin to our approach (as referenced in line 212). For training time, PGDM required approximately 5 hours of training on a single-tracked trajectory, while our method completed supervised learning and fine-tuning in less than 60 minutes. For tracking errors, both methods reported similar tracking errors, indicating comparable accuracy levels. These results highlight the efficiency of our method without compromising accuracy, offering a promising pre-training with a model-based approach.
>
> |       |   PGDM   |  Ours |
> | :-------:  | :---: | :---: |
> | Ball      | 284 $\pm$ 5 $\ $2.4$\degree$ $\quad\ $  | 55 $\pm$ 4 $\ $ 2.6$\degree$ $\quad\ $
> | Cube   |  292 $\pm$ 3 $\ $4.1$\degree$$\quad\ $  | 59 $\pm$ 2 $\ $ 3.8$\degree$$\quad\ $
> | Capsule   |  301$\pm$6 $\ $9.3$\degree$$\quad\ $  | 57 $\pm$ 2 $\ $ 7.7$\degree$$\quad\ $
>
> Baseline comparisons on contact-aware manipulation policy. The first element in each cell is the mean/variance for the training time (m); the second is the difference between the target and final object rotation (°)
>
>
> **Q5. How well will this approach generalize to more challenging manipulation tasks? For example, many manipulation tasks (e.g. pouring, insertion) are not simply about minimizing the pose error**
>
> While the present work concentrates on object-tracking tasks within a learning-from-demonstration framework, extending the model-based approach to more intricate manipulation tasks stands as a pivotal future endeavor on our roadmap. Recent studies have showcased the potential of utilizing differentiable simulations in complex manipulation tasks [2,3]. However, the optimization of actions with a sparse loss function over contact continues to pose a challenge. Our planning component addresses this issue by explicitly inferring contact points and optimizing for both contact transition and wrench execution separately. This innovation not only mitigates existing challenges but also lays the groundwork for potential generalization to other demanding tasks. One possible solution to more challenging tasks is to infer object trajectory for the task and then optimize manipulation trajectories as we proposed. We are optimistic about the applicability of this concept and plan to validate our hypothesis in future research.
>
> **Q6. How long does your approach take to train on new objects?**
>
> In complex in-hand manipulation tasks, our approach takes 73 seconds in the planning phase on average. During supervised learning, convergence is achieved in approximately 9.3 minutes. The fine-tuning takes an average of 50.1 minutes, utilizing 20 CPU cores and an RTX 3090 GPU.

---

> ### Author Response · Authors · 2023-08-13
> **Authors' Response to Reviewer jZUA (Part 3)**
>
> **Q7. How are the contact points chosen? Are they manually specified for each object, or are they somehow learned from the reconstruction?**
>
> The process of inferring contact points is achieved through enumeration, as detailed in lines 114-135. The algorithm operates in the following stages:
>
> 1. Determine the desired object transformation wrench: Initially, the desired object transformation wrench (at the object's center of mass) is derived from the tracked object trajectories with a PD controller.
> 2. Scan possible contact points: The algorithm scans all potential contact points on the object's surface, filtering those capable of generating the desired object transformation wrench.
> 3. Identify executable contact points: Further refinement is done to pinpoint executable contact points that stabilize the object during the contact transition.
>
> **We hope that our responses address your concerns/questions. Thank you so much for reviewing our paper and providing constructive feedback!**
>
> [1] Dasari et al. Learning Dexterous Manipulation from Exemplar Object Trajectories and Pre-Grasps. 2023 ICRA.
>
> [2] T. Pang, H. J. T. Suh, L. Yang, and R. Tedrake. Global planning for contact-rich manipulation via local smoothing of quasi-dynamic contact models, 2022.
>
> [3] R. Antonova, J. Yang, K. M. Jatavallabhula, and J. Bohg. Rethinking optimization with differentiable simulation from a global perspective. In 6th Annual Conference on Robot Learning, 2022.

---

### Official Review · Reviewer_YF65 · 2023-07-19

**Confidence:** 3
**Originality:** Good
**Technical Quality:** Very Good
**Clarity Of Presentation:** Very Good
**Impact:** 3

**Recommendation:**

Weak Accept: I recommend accepting the paper, but will not argue for my recommendation if the majority of other reviewers have a different opinion.

**Review:**

Strengths
- It proposes a very comprehensive system for learning from video on the challenging manipulation task, such as in-hand reorientation. It's really dense, since almost every step is tough, i.e., estimating full object shape from video, plan and optimize contact actions. It's really impressive that the authors are able to put them together and make it work.
- The proposed hierarchical policy is very interesting. It integrates high-level contact sequence planning and low-level contact-point localization and contact-force optimization using differentiable simulation. The results look very promising.

Weaknesses
- Although in the related works section the authors cite a lot of works, the positioning and contribution of the paper isn't very clear to me. For example, it cites many papers under "Differentiable Simulation and Rendering in Robotic Manipulation", but it doesn't state clearly that comparing to prior works, what's the most significant difference. Instead of citing all papers at the same time, the author should select the most relevant ones and explain the key differences.  My feeling is that object tracking and reconstruction through differentiable rendering is a well-studied field. If the proposed pipeline is built off prior works, the author should cite it in the method section. If the proposed method contains new insights, the authors should also make it clear. Same thing for using differentable simulation for trajectory optimization.

**Quality Of The Limitations Section:**

Limitations are addressed clearly

**Questions For Rebuttal:**

- With the proposed self-supervised reconstruction method, we're able to recover the full shape of the object. However, I'm wondering how you obtain the physical parameters, such as friction and mass.
- Why do you need to recover the texture of the object, since it's not helpful for manipulation task. Can't you just render the point cloud and recover the shape?
- In algo 1, line 5, how does the SelectNode work? Do you adopt any heuristic function to make the planning more efficient?
- Questions regarding the closed-loop policy:
    - Does it take both current pose and goal pose of the object as input? How do you obtain the pose?
    - How do you obtain the contact configurations? Do you also apply the high-level contact planning at test time, or do you assume it's known?

- Typos:
    - Line 52: prepossess -> pre-process

**Robotics Focus:**

Sufficient demonstration on hardware

**Summary Of Paper:**

The paper presents a framwork for model-based Learning from Demonstration (LfD) in the context of contact-rich manipulation. The proposed framework contributes to the following aspects:
1. The usage of differentiable rendering to track and reconstruct object poses in a self-supervised manner.
2. A hierarchical planning framework that employs differentiable simulations to generate long-horizon contact actions.
3. A sim-to-real neural policy that is trained by domain randomization.

The authors demonstrate the effectiveness of the method on several manipulation tasks, including pulling, pushing and in-hand reorientation. Results in both simulation and real world look very promising.

**Summary Of Recommendation:**

The paper proposes a novel framework with carefully designed perception and planning component, to solve a very challenging task, learning contact-rich manipulation from video.  The presented results are very impressive. Hence, I recommend accepting the paper.

---
Aftering rebuttal, my rating remains unchanged. I agree that there isn't much novelty in the individual sub-system, but I appreciate the system-level contribution of putting all together. Reconstruction by differentiable rendering has been a hype in the vision community for a while, but it's interesting to see the reconstructed object is actually good enough for manipulation. I believe the community can be benefited from reading this paper.

---

> ### Author Response · Authors · 2023-08-13
> **Authors' Response to Reviewer YF65**
>
> Thank you for the positive rating and evaluations. We appreciate the reviewer’s constructive suggestions/questions/comments for further clarifying and improving our work.
>
> **Below, we respectively address your questions/comments**
>
> **Some questions are answered in the general response.**
>
> **Q1. Why do you need to recover the texture of the object since it's not helpful for manipulation task? Can't you just render the point cloud and recover the shape?**
>
> Our hypothesis is that texture of an object might be beneficial to estimate object poses.
>
> **Q2. In algo 1, line 5, how does the SelectNode work? Do you adopt any heuristic function to make the planning more efficient?**
>
> We employ the A* search algorithm for node selection during expansion. Adopting the same notations presented on [Wikipedia](https://en.wikipedia.org/wiki/A*_search_algorithm):
> Cost from the start (g): This is set to the number of steps taken so far. Both ContactLoc and OptWrench have a step size of 1.
> Heuristic cost (h): This is designed as the SE3 difference between the current node and the target. Given that the robot cannot transition an object by 1m or 1rad in a single move, the consistency of the heuristic is thereby ensured.
>
> **Q3. Questions regarding the closed-loop policy: Does it take both the current pose and goal pose of the object as input?**
>
> Our closed-loop policy is designed to operate without taking the goal poses of the object as input. Instead, it is specifically trained for each individual manipulation trajectory. The challenge of generalizing this approach to various objects and trajectories is an exciting avenue we plan to explore in our future work.
>
> **Q4. How do you obtain the pose?**
>
> During deployment, the object's pose is continuously monitored using the pose estimation module, as detailed in Sec. 2.1.
>
> **Q5. How do you obtain the contact configurations? Do you also apply the high-level contact planning at test time, or do you assume it's known?**
>
> The contact configuration is encoded from the robot's joint angles and the object's pose during testing. We have corrected a typo in line 201, clarifying that the network's input consists of the object's pose and joint angles rather than the explicit contact configuration. Moreover, the closed-loop policy does not require contact planning at the test time; it is assumed to be known and imitated during the training.
>
> **We hope that our responses are well addressing your concerns/questions. Thank you so much for reviewing our paper and providing constructive feedbacks!**

---

### Official Review · Reviewer_Qw95 · 2023-07-19

**Confidence:** 4
**Originality:** Good
**Technical Quality:** Good
**Clarity Of Presentation:** Good
**Impact:** 3

**Recommendation:**

Weak Accept: I recommend accepting the paper, but will not argue for my recommendation if the majority of other reviewers have a different opinion.

**Review:**

## Strengths

The approach to involve a simulator in the visual LfD methodology is very interesting. The learning of the gemetric and visual shape without requiring the camera poses appears to work quite well, which is demonstrated through extensive experiments in the supplemental material (although some problems remain when the object is occluded throughout the video). The authors perform extensive experiments to demonstrate many parts of the proposed pipeline.

## Weaknesses

The object reconstruction appears to have issues when parts of the object are occluded (a common problem in visual footage of manipulation behaviors), as can be seen e.g. in the coffee cup in Figure 2 of the supplemental material. The proposed diffusion model does not seem to help; the approach to combine imagined viewpoints and real images of the object for training the SDF is not explained further in the supplemental material, although it is mentioned in the paper (lines 99-100).

The baselines in the experiments are quite weak, particularly RRT search over the joint configuration space is too inefficient to be considered state of the art. A more capable baseline, such as [PGDM](https://pregrasps.github.io/) that also generates manipulation behaviors given the desired motion of an object, would be much better suited to compare the performance of the proposed framework.

The neural network policy seems to require retraining (including the generation of a training dataset) for every new object and reference video, which limits the real-time applicability to manipulation behavior generation.

The generated behaviors only imitate the motion of the object and completely discard the motion of the human hand manipulating the object. Here it would have been interesting to learn from the way the hand moves more efficient/natural-looking robot hand motions than the one-finger-at-a-time behavior that the current approach generates.

## Minor issues

"Table 1" in line 242 should refer to Figure 3

**Quality Of The Limitations Section:**

Limitations are addressed clearly

**Questions For Rebuttal:**

How is the camera pose resolved? It appears to be set to $\lbrace\hat{\mathcal{P}}^*_t\rbrace$ (line 89), but it is mentioned earlier that these are the *object* poses.

The approach from Sec. 2.2 is difficult to follow. It is unclear to me why the localization of contact points involves the "execution" of the pose transition wrench(?) when the optimization of the contact wrench actually happens in another function (OptWrench). More details are needed on the possible combinations of contact points: do these point candidate correspond to the mesh vertices or how are they generated? Why is a wrench computed during the point localization step when the contact filtering (for which details are missing) appears to happen solely based on whether the contact wrench direction (so PD gains do not matter?) is inside the contact friction cone?

What are the "stationary constraints" mentioned in line 126-127?

It is mentioned that the analytical contact wrench from ContactLoc is used to initialize the wrench optimization in OptWrench (line 158-159). In Algorithm 1 the first iteration starts with OptWrench however without previously running ContactLoc, so what initialization is used here?

Are the inertia of the object considered in the OptWrench step? How are the inertial properties estimated from the observed object?

The objects reconstructed in the vision-only experiments (Figure 2 in supplemental material, or the videos from the website) are significantly more complex than the box or sphere experiments in the motion generation experiments. I wonder what the limitations are to apply this pipeline to such geometry or whether there is a preference to be limited to simple convex shapes?

As mentioned under "weaknesses" of this review, it is advisable to improve the performance comparison from Figure 4 to include a baseline that is actually state-of-the-art in manipulation behavior generation, such as [PGDM](https://pregrasps.github.io/), or a trajectory optimization approach that leverages the simulator gradients (e.g. iLQR, CIO) or not (e.g. MPPI).

**Robotics Focus:**

Sufficient demonstration on hardware

**Summary Of Paper:**

This paper introduces a visual learning from demonstrations (LfD) framework that first reconstructs and tracks a rigid object being manipulated in an RGB video. Given the shape and motion of the object, a set of contact points and wrenches are found to realize the reference motion in a simulator. Given the planned contact trajectory, the joint torques are recovered via the robot's kinematic Jacobian and a neural network policy is trained from perturbed data to improve the sim2real transfer. Real robot experiments on an Allegro hand are performed to demonstrate the pipeline on convex objects.

**Summary Of Recommendation:**

The proposed visual LfD pipeline is novel and combined various techniques from differentiable rendering and simulation to tackle the problem of learning from manipulation videos. The results are overall convincing, albeit the considered objects are simple and the considered baselines are not state of the art. The clarity of the description of the technical details (Sec. 2.1 and 2.2.) needs to be improved.

---

> ### Author Response · Authors · 2023-08-13
> **Authors' Response to Reviewer Qw95 (Part 1)**
>
> Thank you for the positive rating and evaluations. We appreciate the reviewer’s constructive suggestions/questions/comments for further clarifying and improving our work.
>
> **Below, we respectively address your questions/comments**
>
> **Q1. How is the camera pose resolved? It appears to be set in line 89, but it is mentioned earlier that these are the object poses.**
>
> During the hierarchical pose estimation process, we optimize the object's pose within the camera frame, referred to as the "object pose." Subsequently, in the SDF optimization, we utilize the relative pose of the camera to the object, termed the "camera pose." Therefore, the correct notation should be $\{\hat{\mathcal{P}}_t^{-*}\}$ as indicated in line 89. We have identified and corrected this typo in our paper.
>
> **Q2. The approach from Sec. 2.2 is difficult to follow. It is unclear to me why the localization of contact points involves the "execution" of the pose transition wrench(?) when the optimization of the contact wrench actually happens in another function (OptWrench). Why is a wrench computed during the point localization step when the contact filtering (for which details are missing) appears to happen solely based on whether the contact wrench direction (so PD gains do not matter?) is inside the contact friction cone?**
>
> We have modified the paper for a clearer presentation. In the determination of new contact points, we introduced two types of wrenches: desired object transformation wrench $\mathcal{W}$ and contact wrench $\mathcal{W}^p$.
>
> 1. Desired object transformation wrench $\mathcal{W}$: Located at the object’s center of mass and represents a desired wrench to transform the object between poses. It cannot be executed by the robot and is computed by the PD controller.
>
> 2. Contact wrench $\mathcal{W}^p$: Positioned at the contact point $p$, this wrench is executable by making contact on the surface. A mapping exists to transform the surface contact wrench to a CoM wrench, as described in Sec. 1.2 in the supplementary material.
>
> During contact point localization, the goal is to identify contact points $p$ capable of producing the desired object transformation wrench $\mathcal{W}$ with contact wrenches within its friction cone. This process is mathematically illustrated in Eq. 7 in the supplementary material, where the l2-norm represents the objective that the contact wrench $\mathcal{W}^p$ should produce $\mathcal{W}$. The PD gains used to compute $\mathcal{W}$ are manually tuned, with values provided in line 117 of the supplementary material.
>
> **Q3. More details are needed on the possible combinations of contact points: do these point candidates correspond to the mesh vertices, or how are they generated?**
>
> In our implementation, points are uniformly sampled from the object’s mesh with trimesh’s sample_surface_even function. These points lie on the surface of the mesh.
>
> **Q4. What are the "stationary constraints" mentioned in lines 126-127?**
>
> The stationary constraints mandate that the object remains immobile during contact transitions. To elucidate, in the context of in-hand manipulation tasks: when one finger relocates to a new contact point, the remaining fingers must securely hold the object in its position without letting it fall.
>
> **Q5. It is mentioned that the analytical contact wrench from ContactLoc is used to initialize the wrench optimization in OptWrench (line 158-159). In Algorithm 1 the first iteration starts with OptWrench however without previously running ContactLoc, so what initialization is used here?**
>
> We appreciate the reviewer's thorough feedback. For the initial run, we set the OptWrench using a unit contact wrench that aligns with the contact's normal direction. To elaborate: given the normal direction of a contact point $p$ is $n$, the initial $\mathcal{W}^p$ is initialized to $[n^1, n^2, n^3, 0, 0, 0]^T$, where $n=[n^1, n^2, n^3]^T$.
>
> **Q6. Are the inertia of the object considered in the OptWrench step? How are the inertial properties estimated from the observed object?**
>
> Indeed, we incorporate the object's inertia into the simulation. In our work, the inertia is positioned at the geometrical center of the reconstructed mesh. We also provided details on how we leverage physical parameters in Q6 in the general response.

---

> ### Author Response · Authors · 2023-08-13
> **Authors' Response to Reviewer Qw95 (Part 2)**
>
> **Q7. The objects reconstructed in the vision-only experiments (Figure 2 in supplemental material or the videos from the website) are significantly more complex than the box or sphere experiments in the motion generation experiments. I wonder what the limitations are to applying this pipeline to such geometry or whether there is a preference to be limited to simple convex shapes.**
>
> On our website, we have posed the object shape used in our experiments. The objects vary significantly in geometry. For the manipulation tasks, we opted for simple shapes, following the precedent set by previous works [8]. This choice also accommodates the numerical stability of the differentiable simulation, where complex mesh-mesh contacts can sometimes prove unstable. To further explore the capabilities of our approach, we conducted additional experiments to test the closed-loop policy with more intricate geometries; the results are discussed in Q3 in the general response.
>
> **Q8. As mentioned under "weaknesses" of this review, it is advisable to improve the performance comparison from Figure 4 to include a baseline that is actually state-of-the-art in manipulation behaviour generation, such as PGDM, or a trajectory optimization approach that leverages the simulator gradients (e.g., iLQR, CIO) or not (e.g. MPPI).**
>
> Thanks for the suggestion. We conducted more baseline comparisons as the reviewer suggested over PGDM and iLQR; the details are illustrated in Q5 in the general response.
>
> **Q9. The object reconstruction appears to have issues when parts of the object are occluded (a common problem in visual footage of manipulation behaviors), as can be seen e.g. in the coffee cup in Figure 2 of the supplemental material. The proposed diffusion model does not seem to help; the approach to combine imagined viewpoints and real images of the object for training the SDF is not explained further in the supplemental material, although it is mentioned in the paper (lines 99-100).**
>
> We thank the reviewer for the question. We have added experiments for the diffusion
> model. Please see Q4 in the general response for more details.
>
> [8] Global Planning for Contact-Rich Manipulation via Local Smoothing of Quasi-dynamic Contact Models. Tao Pang, H.J. Terry Suh, Lujie Yang, Russ Tedrake. arXiv Preprint.

---

### Official Review · Reviewer_kobW · 2023-07-21

**Confidence:** 4
**Originality:** Fair
**Technical Quality:** Good
**Clarity Of Presentation:** Very Good
**Impact:** 3

**Recommendation:**

Weak Accept: I recommend accepting the paper, but will not argue for my recommendation if the majority of other reviewers have a different opinion.

**Review:**

# Originality

This paper proposes a reasonable system for watching a demonstration of an object being manipulated, estimating its geometry, and using that estimated geometry and a model-based planner/controller to control a robot to reorient the same object. These two sub-components are quite well-studied areas (3D reconstruction from motion, and contact-rich in-hand manipulation), so I wouldn’t consider the modules themselves novel (as I’ve seen differentiable rendering for reconstruction and contact-aware planning in prior work). However, getting a full manipulation system to work based on these two components is something I haven’t really seen before, and it’s a good system application of advances in visual reconstruction.

# Quality

The paper is high quality in terms of execution. The experiments seem thorough, the description of the algorithms is complete and comprehensive (with a few minor exceptions, highlighted in questions).

However, several of the design decisions seemed somewhat arbitrary:
The multi-step reconstruction procedure seems quite empirical, but no ablations were really provided for design choices. Additionally, the baseline method provided is not very strong - there are many other competitive algorithms for this sort of reconstruction - even with differentiable rendering, like the entire NeRF lineage of work - which isn’t compared against. That’s okay: but there isn’t much empirical justification for the design of their method.
For the MLP trained to approximate the planning process, why was REINFORCE used instead of supervised learning? Especially if the model itself is differentiable.

# Clarity

The paper is reasonably well-written. However several aspects were unclear:
How is mass estimated / taken into account for contact planning?
What is the alignment procedure for novel objects? Details were not clear in the supplement.
Why was diffusion necessary?
Line 99: how are synthetic views used? This was especially unclear and the supplement did not clear things up.

# Significance

The technical aspects of this paper are interesting (particularly the test-time optimization), but I suppose my biggest question is whether this method’s main contribution (reconstruction) has benefits compared to other structure-from-motion work? Each of the two major components seem related to existing work.

# Relevance

While I believe that the paper is high-quality, I don’t believe that this paper is a good fit for this venue. The two primary components of the method are a visual estimation pipeline (which uses a differentiable renderer + optimization, and incorporates no learning components), and a model-based planning and control framework which is purely physics-based. Because these components rely on their quite rigid assumptions to succeed, in my mind there is not a clear path to extend this framework to diverse classes of objects with different material properties, generalize across demonstrations, or to manipulate tasks with dynamics which are much harder to model explicitly. I suppose in principle you could replace each of these modules with a learned version which can represent their functions more flexibly, but that would be an entirely different paper.

Additionally - and I’m not totally sure about this point - but it feels a bit strange to call this method “Learning from Demonstrations”. Typically, that label is used when a policy itself is learned (could be a visual policy); but as far as I can tell, there is no learning at the policy level (except for a minor policy learning component for real-time execution) - it’s purely analytical. There’s something of an argument to be made that the visual component is learning from a demonstration, although one might quibble over whether optimizing a representation for a specific example is learning.

# Limitations
The biggest unaddressed limitation is generalization to novel objects with different geometry at test time. The authors attempt to address this earlier in the text by saying that you can just align that object to a given demonstration, but provide no details on how to do this (and I can think of many scenarios where such an alignment wouldn’t work).

**Quality Of The Limitations Section:**

Additional details required

**Questions For Rebuttal:**

Can you elaborate on how you would actually use this system at test-time? In terms of specifying inputs/outputs after watching a demo.

Can you elaborate more on how you align novel objects to existing demonstrations?

**Robotics Focus:**

Sufficient demonstration on hardware

**Summary Of Paper:**

In this work, the authors propose a method to reconstruct the geometry of an object from RGB demonstrations, which they use in a downstream manipulation policy which uses model-based control and planning to plan contact-rich manipulation. The authors demonstrate that their reconstruction approach is effective in simulated and real experiments, and that their planning approach can successfully manipulate objects in several real-world experiments.

**Summary Of Recommendation:**

Overall, I am impressed with the quality of the system that was constructed and demonstrated: going from visual demonstrations directly to contact-rich control is not an easy feat. However, I don’t think this paper is appropriate for this venue - the typical “learning” components which the community is interested in are absent, and this approach does not demonstrate abilities/behavior which the community will find relevant (i.e. generalization, flexibility, etc.). The community might be interested in the subsystems themselves at a technical level, but I feel that this work is a better fit for a pure robotics venue.

---
Update: based on the rebuttal and discussion with other reviewers, I now believe that this is more appropriate for the learning community than I initially believed. So I have updated from Strong Reject to Weak Accept. See comment for details.

---

> ### Author Response · Authors · 2023-08-13
> **Authors' Response to Reviewer kobW**
>
> Thank you for the rating and evaluations. We appreciate the reviewer’s constructive suggestions/questions/comments for further clarifying and improving our work.
>
> **Below, we respectively address your questions/comments**
>
> **Q1. Ablations and comparisons.**
>
> We added more experiments to compare our method, both perception and planning parts, to existing end-to-end methods. We also provided ablation studies to the hierarchical reconstruction and diffusion model, as the reviewer suggested. Please see Q2, Q3, Q4, Q5 in the general response for details.
>
> **Q2. Unclear details of the method.**
>
> We modified the supplementary material on our website and provided details on how we leverage physical parameters in Q6 in the general response.
>
> **Q3. Scope of learning from demonstration**
>
> We justified the contribution of our work and relevance to the community in Q1 in the general response.
>
> **Q4. Can you elaborate on how you would actually use this system at test time? In terms of specifying inputs/outputs after watching a demo.**
>
> During test time, the algorithm proceeds through the following stages.
> 1. Object extraction: Initially, objects of interest are extracted using a segmentation network, i.e., SAM.
> 2. Pose and shape estimation: The masked RGB images are then fed into the pose and shape estimation module. This outputs the object mesh, and 6D poses for each frame.
> 3. Contact-aware planning: The tracked pose and mesh are input into the contact-aware planning stage. This generates canonical manipulation trajectories with injected noises.
> 4. Closed-loop policy training: The manipulation trajectories are utilized to train a neural network. The network is designed to imitate the video demonstrations of the manipulation task. The neural network undergoes additional refinement to enhance its ability to replicate the demonstrated manipulations accurately. Notably, the closed-loop policy developed through this process can be deployed to novel objects. Detailed experiments and discussions supporting this capability are available in Q3 of the general response.
>
>
> **Q5. Can you elaborate more on how you align novel objects to existing demonstrations?**
>
> The alignment is solved as a pose estimation problem: $\arg\min_{\mathcal{P}} \textrm{d}(\hat{\mathcal{P}}_0^{* *}\hat{\mathcal{O}}^{**}, \mathcal{P}\mathcal{O})$.
>
> $\textrm{d}$ is the chamfer distance between two meshes, $\hat{\mathcal{P}}_0^{**}$ is the pose of the reconstructed mesh at the first frame, $\hat{\mathcal{O}}^{* *}$ is the reconstructed mesh, $\mathcal{P}$ is a transformation to the novel object, and $\mathcal{O}$ is the novel object.
>
> After solving the optimization, we apply $\mathcal{P}^{-1}$ to tracked object pose $\hat{\mathcal{P}}_{t}^{**}$ to obtain the pose of the novel object for the contact-aware manipulation $\mathcal{P}_t$.
>
> **Q6. What is the alignment procedure for novel objects?**
>
> Given a novel object, our default procedure is to sample point clouds on the novel objects and our reconstructed object and perform point cloud registration to align these two objects. If the users want to specify a particular starting pose for the object, we will skip the alignment.
>
> Most important, the results of the alignment procedure have little effect on our pipeline. In the first time step, during the contact point localization, all possible combinations of contact points are enumerated to ensure that there are feasible contact points. Then our proposed pipeline (for the manipulation policy generation) focused on generating feasible contact wrenches to transform objects between two consecutive poses.
>
> **We hope that our responses are well addressing your concerns/questions. Thank you so much for reviewing our paper and providing constructive feedbacks!**

---

### Author Response · Authors · 2023-08-12
**Authors' General Response to Reviewers/AC (Part 1)**

We appreciate the time and efforts of all the reviewers and AC in reviewing our submission. We sincerely thank all the reviewers and AC for the constructive feedback and suggestions for further clarifying and improving our work.

Firstly, we are glad and thankful to know that the reviewers and AC are encouraged by the novelty, high quality, and/or importance of our proposed method/framework. We really hope that our work can make reasonable and useful contributions and possibly may inspire future research on investigating how to use techniques of differentiable simulation and rendering to tackle the learning from demonstration problem in a self-supervised manner and make it generate large-scale robot skills at low costs.

In this comment, we focus on addressing the major/shared concerns and questions from the reviewers and the AC.

**Q1. Scope of Learning from Demonstrations (Reviewer kobW)**

The problem of Learning from demonstrations (LfD) refers to having “robots learn a task by watching the task being performed by a human or by another robot. The robot can either mimic the motion of the demonstrator or learn how the demonstrator acts, reacts, and handle eros in many situations”[1].

The problem formulation of LfD exists long before the era of Deep Learning.
Various approaches without neural networks (e.g., DMP[2,3]) have been proposed to tackle the LfD problem before Deep Learning. A survey [4] of LfD in 2009 is referred to below. The formulation of LfD is independent of whether using neural networks as policy approximators as long as the robots can imitate the demonstrated behavior autonomously without manual programming.

Our Diff-LfD pipeline would first estimate the object shapes and poses and then figure out how to generate executable robot action to mimic the same pose transformation as the demonstration (in the differentiable simulator) and then train a neural network to perform closed-loop policy to accomplish the task in the real robot.

From a high-level perspective, the input to our framework is a demonstrated video. Our pipeline outputs a policy for real robots to accomplish the task to mimic the demonstrated task. Thus, our inputs and outputs fall into the scope of LfD.

[1] Robot Learning From Demonstration. Christopher G. Atkeson and Stefan Schaal. ICML, 1997.

[2] Dynamical Movement Primitives: Learning Attractor Models for Motor Behaviors. Auke Jan Ijspeertal,Jun Nakanishi,Heiko Hoffmann,Peter Pastor,Stefan Schaal. Neural Computation, 2013.

[3] Learning rhythmic movements by demonstration using nonlinear oscillators.  Auke Jan Ijspeertal,Jun Nakanishi, Stefan Schaal. IROS, 2002.

[4] A survey of robot learning from demonstration. Brenna D. Argall, Sonia Chernova, Manuela Veloso, Brett Browning. Robotics and Autonomous Systems. Volume 57, Issue 5,
2009.

**Q2. Comparison to NeRF-related works (Reviewer kobW, YF65, jZUA)**

We will illustrate the limitations of the current NeRF series work in our problem setting and the advantages of our work over the NeRF series work from both analytical and experimental aspects,  which helps demonstrate the difficulty of our task and the effectiveness of our proposed visual framework.

Our work has a different task setting than the NeRF approach. In our problem setting (only the perception part of our pipeline): the input is a single-object RGB video, where objects in the videos are rotated and translated, and the expected output is the object reconstruction and the associated object poses in each frame. Most NeRF works take multiple poses of a static object (without large translation and rotation) as input and produce a rendered image of a new view or a reconstruction of an object's surface. Thus, to the best of our knowledge, no current work in the NeRF family accomplishes this task.

Even considering the relativity of camera and object motion, converting our task to a setting where the object is static and the camera is moving still fails the NeRF family of works for the following reasons: Most of the NeRF works rely on the initialization of the camera pose provided by COLMAP, while in our task scene, the background hardly changes and the number of frames is small, so COLMAP will fail to estimate the object poses, resulting in the inability to complete the reconstruction of the object surface.

However, suppose we don't consider the quality of the object surface reconstruction and only consider the pose estimation without an object model. In that case, we also note that there are some recent works, such as Nope-Nerf(CVPR2023), and compare it with our method.  Results are available on our [website](https://sites.google.com/view/diff-lfd). We observe that in more than half of the test samples (5/9), Nope-Nerf fails to converge or converges to a white background, producing blank rendered videos. The rest test examples that converged during training (4/9) produced apparently incorrect pose estimation and reconstruction results.

---

### Author Response · Authors · 2023-08-12
**Authors' General Response to Reviewers/AC (Part 2)**

**Q3. Generalization of LfD (Reviewer kobW, Reviewer Qw95)**

Given a demonstrated video, our proposed framework can work with similar but different objects.  We have conducted additional experiments to demonstrate the generalization capabilities of the closed-loop policy (a trained neural network model). We directly test the trained neural network policy to novel objects without retraining.

The policy was previoulsy trained to imitate human actions in rotating a cube. The results of the difference between the target and final object rotation errors are Cylinder (3.8$\degree$), Ball (2.4$\degree$), Lemon (6.3$\degree$), and Avocado (5.9$\degree$), which further underscore the ability of our closed-loop policy to generalize across similar but distinct geometries. The result is also available on the project [website](https://sites.google.com/view/diff-lfd).

**Q4. Diffusion Model ( Reviewer kobW, Qw95, jZUA)**

We perform a comparison experiment with and without the diffusion model and post it on our project  [website](https://sites.google.com/view/diff-lfd). The hallucinated/unseen side of the object looks more reasonable using the diffusion model.

We propose to utilize the pre-trained Zero 1-to-3 diffusion model in our approach. Zero 1-to-3 is a specialized framework designed for novel view synthesis from a single RGB image. During the inference phase, the Zero 1-to-3 diffusion model takes the input view and a relative viewpoint as conditional information, synthesizing the corresponding novel view. It's important to note that we do not train diffusion models within our pipeline. Instead, we leverage these pre-trained diffusion models, honed on extensive datasets, to provide supervision on unseen views. This approach is particularly applied during the reconstruction and pose estimation phases of our methodology.


**Q5. Comparison to baselines (Reviewer kobW, Qw95, jZUA)**

For the perception part, we add ablation studies to the perception module to justify the design of pose estimation and mesh reconstruction reported in the supplementary material (Page 5, Table 1).

For the planning part, the MLP is first trained with planned trajectories and fine-tuned with REINFORCE. We recognize the potential of incorporating supervised learning during the fine-tuning phase, presenting a promising avenue for future research. However, as highlighted in [8] and corroborated by our experiments, optimizing the entire trajectory solely through gradient descent poses challenges. This is primarily because of the contact discontinuity and the noisy gradients. The contact dynamics are discontinuous, and the complex physical simulations often result in noisy gradient signals, making optimization more challenging.

In response to the reviewer's suggestion, we incorporated comparisons of our planning approach with both PGDM [5] and iLQR. The results are tabulated below. While PGDM exhibits the shortest inference time, it's essential to note that its manipulation policy underwent training for approximately 5 hours on a single-tracked trajectory using PPO with default hyperparameters. The tracking error from PGDM is comparable to our approach, indicating similar accuracy levels. iLQR's planning time aligns closely with ours. However, its inferior performance might be attributed to its optimization of only the first-order approximation.

|       |    $\qquad$ RRT   | $\quad\ $ CITO | $\quad \ \ $ PGDM | $\quad \ \ $ iLQR | $\quad \ \ $ Ours |
| :-------:  | :---: | :---: | :---: | :---: | :---: |
| Ball      | 122 $\pm$ 20 $\ $7.2$\degree$ $\quad\ $  | 52 $\pm$ 8 $\ $ 16.7$\degree$ $\quad\ $ | 2.14 $\pm$ 0.4 $\ $2.4$\degree$ $\quad\ $ | 57 $\pm$ 10 $\ $ 11.2$\degree$ $\quad\ $ | 62 $\pm$ 12 $\ $ 2.6$\degree$ |
| Cube   |  136 $\pm$ 16 $\ $9.0$\degree$$\quad\ $  | 60 $\pm$ 7 $\ $ 18.5$\degree$$\quad\ $ | 2.16 $\pm$ 0.3 $\ $4.1$\degree$$\quad\ $ | 70 $\pm$ 19 $\ $ 13.3$\degree$$\quad\ $ | 78 $\pm$ 16 $\ $ 3.8$\degree$ |
| Capsule   |  127 $\pm$ 24 $\ $8.4$\degree$$\quad\ $  | 63 $\pm$ 4 $\ $ 15.2$\degree$$\quad\ $ | 2.18 $\pm$ 0.3 $\ $9.3$\degree$$\quad\ $ | 80 $\pm$ 6 $\ $ 12.2$\degree$$\quad\ $ | 81 $\pm$ 8 $\ $ 7.7$\degree$ |

Baseline comparisons on contact-aware manipulation policy. The first element in each cell is the mean/variance for the computation time ($s$); the second is the difference between the target and final object rotation ($\degree$).

[5] Learning Dexterous Manipulation from Exemplar Object Trajectories and Pre-Grasps. Sudeep Dasari, Abhinav Gupta, Vikash Kumar, ICRA, 2023.

[8] Global Planning for Contact-Rich Manipulation via Local Smoothing of Quasi-dynamic Contact Models. Tao Pang, H.J. Terry Suh, Lujie Yang, Russ Tedrake. arXiv Preprint.

---

### Author Response · Authors · 2023-08-12
**Authors' General Response to Reviewers/AC (Part 3)**

**Q6. Physical Parameters ( Reviewer kobW, jZUA)**

We do not derive the physical parameters, such as mass, from observations. Instead, for each planning trial, they are sampled from a uniform distribution. In real-world experiments, we employ these randomized physical parameters during planning to amass a dataset, which is then used to train the closed-loop policy.


**Q7. Our contributions (Reviewer jZUA)**

For the systematic level, we proposed a novel model-based framework for learning from demonstration to empower robots to mimic visual demonstrations. The framework utilizes the “comparing sim-real and then improving” fashion leveraging the differentiable simulation and rendering, thus is in a self-supervision manner, which has the potential to generate robot skills at a big scale.

In order to achieve that, we proposed a perception module that takes object shapes and poses from a demonstrated RGB video without camera poses, and we have demonstrated that our proposed pipeline can deal with a large variety of object geometry (see the video in the website).

To generate the policy efficiently, we proposed a model-based policy generation pipeline combining the differentiable simulation and high-level planning. We propose to infer the contact point under the kinematic constraints and generate a low-level policy with the differentiable simulation. With the contact point as the intermediate step, we proposed high-level planning to overcome the local optimal issue since the differentiable operation provides only local information. The hierarchical policy generation outperforms existing approaches by a large margin (see the results in the mainpaper).

**We also addressed other comments/questions/issues raised up by each specific reviewer by directly replying to each reviewer.**

We have revised the main paper, the supplementary document, and the video, to incorporate major changes. We would be very happy to incorporate any other requested changes in the final paper if accepted.

Thanks again for your valuable reviews/comments!

Best regards, Authors

---

### Decision · Program_Chairs · 2023-08-30

**Decision:**

Accept (Oral)

**Comment:**

This paper introduces a pipeline for learning from demonstration that incorporates a number of elements, self-supervised pose and shape estimation, and contact sequence generation, building on differentiable rendering, with contact point and force optimisation. Reviewers initially found this a polarising paper, with two leaning to accept and two in favour of rejecting. The primary concerns were around the system's level nature of this paper and potential fit for CoRL. Reviewers felt that the overall pipeline made this an impressive systems paper, with convincing results, but that subsystems are relatively standard components, and that there is limited theoretical novelty or contribution to robot learning in these individual subsystems. There were also concerns that the policy is generated via optimisation/planning, not learned.

The authors provided a comprehensive and detailed rebuttal, introducing a number of additional experiments. The authors argued that the framework as a whole is novel, and that inference of the state and contact points so as to optimise a policy with respect to a simulation to reproduce these is still learning from demonstration. Reviewers relt that the system and additional experiments helped motivated the fit for CoRL.

I do not think that there is no place for systems papers at CoRL (historically we have given awards to the best of these). Although the idea has been broadly spoken about as a natural route forward, as mentioned by reviewers, we have not yet seen many differentiable rendering and simulation based planning pipelines working on real world manipulation tasks like these yet, so think that this is a contribution that warrants publication. I recommend that the authors update their paper to incorporate their rebuttal responses and additional experiments, which have undoubtedly strengthened the paper.